# Understanding COVID Vaccination and Its Implication in Cancer Patients’ Imaging of Lymph Nodes by PET-CT

**DOI:** 10.3390/diagnostics12092163

**Published:** 2022-09-06

**Authors:** Laurentia Nicoleta Gales, Silvia Brotea-Mosoiu, Oana Gabriela Trifanescu, Alexandra Maria Lazar, Mirela Gherghe

**Affiliations:** 1Oncology Department, Institute of Oncology “Prof. Dr. Alexandru Trestioreanu”, 022328 Bucharest, Romania; 2Oncology Department, University of Medicine and Pharmacy “Carol Davila” Bucharest, 050474 Bucharest, Romania; 3Nuclear Medicine Department, Institute of Oncology “Prof. Dr. Alexandru Trestioreanu”, 022328 Bucharest, Romania; 4Nuclear Medicine Department, University of Medicine and Pharmacy “Carol Davila” Bucharest, 050474 Bucharest, Romania

**Keywords:** COVID-19, ^18^F-FDG PET-CT, mRNA vaccine, cancer management

## Abstract

(1) Background: The appearance of enlarged lymph nodes on imaging adds another layer of complexity to the differential diagnosis of disease progression versus immune response to COVID-19 vaccines. Our aim was to find an optimal timing between the vaccination and the PET-CT scan. (2) Methods: 25 cancer patients with ^18^F-FDG PET-CT evaluations and a history of COVID-19 vaccination between September 2021 and December 2021 were retrospectively analyzed to characterize the lymph nodes related to the time interval from COVID vaccination. (3) Results: All patients presented one or more adenopathies localized in the ipsilateral axilla (96%), ipsilateral cervical area (20%), ipsilateral retropectoral (20%) and pulmonary hilum (8%). The median value of SUVmax was 3.5 ± 0.5. There was a significant indirect correlation between SUVmax and the time passed between the vaccination and the PET CT (Pearson Correlation r = −0.54, *p* = 0.005). There was no significant difference (*p =* 0.19) in the SUVmax value in patients receiving Moderna mRNA-1273 vaccine vs. BNT162b2 mRNA Pfizer vaccine. (4) Conclusions: Lymph node enlargement is commonly seen in patients post-vaccination for COVID-19 and must be differentiated from disease progression. The data from our study strongly suggests that the minimum interval of time between an mRNA vaccine and a PET-CT should be more than six weeks.

## 1. Introduction

In the era of the COVID-19 pandemic, the crucial need for vaccination was discussed from the beginning. Since December 2020, Romania registered a vaccination rate of only 41.9%, almost half of the European mean of 72.4% [1]. Nevertheless, this fact entailed a high rate of mortality among COVID patients, an important percentage of them being cancer patients [2]. Considering this, various recommendations were made, and robust data on the immune response in actively treated cancer patients are emerging or are being published. The latest press release of European Society for Medical Oncology (ESMO) summarized a series of abstracts published for the annual European ESMO Congress in September 2021, all of whom demonstrated the benefits of vaccination and reported a high level of antibodies against COVID-19 after two doses, with the critical need of boosting the effect with a third dose [3,4,5].

When it comes to mass vaccination worldwide, concerns about safety are frequently raised. Side effects are the main reason for the low rate of vaccination stated above in Romania. Many cancer patients avoided the immunization due to concerns related to side effects, while the patients who were vaccinated presented various adverse reactions, raising management difficulties for their attending physicians.

A particular interest in oncology is represented by the immunological response at the site of injection and in the reactive lymphoid tissue surrounding it. Antigens contained in the vaccines interact with the host on a clinical and molecular basis. The first signs of inoculation are clinically referred to as an inflammatory reaction. This phenomenon is compressed by characteristics such as rubor, tumor, calor, and dolor, more commonly known as the Celsus signs. On a microenvironmental and intracellular level, the host responds to the potential pathogen by releasing pro-inflammatory cytokines such as TNF-alpha (tumor necrosis factor alpha), interleukin (IL) 1 and 6, and prostaglandins, by immune cells (macrophages, dendritic cells, monocytes, etc.) located in the nearby vessels of the injection. Mimicking the response to a natural infection, the innate and, furthermore, the adaptive immune system events are activated [6,7,8]. Lymph nodes play a crucial role in shaping the adaptive immune responses, and so the appearance of swollen lymph nodes (or lymphadenopathy) complicates the differential diagnosis between disease progression in solid tumors or hematological malignancies and immune response to COVID-19 vaccine. 

Fluorine-18 Fluorodeoxyglucose Positron Emission Tomography-Computed Tomography (^18^F-FDG PET-CT) is useful for accurate tumor staging in various types of cancer and for monitoring the response to chemotherapy. Accurate nodal staging is one of the great advantages of ^18^F-FDG PET-CT, thus being recommended in several clinical guidelines; however, nonspecific or inflammatory-related ^18^F-FDG uptake in the lymph nodes represents a limitation of this examination [9,10]. There have been reported cases where COVID-19 vaccination mimicked disease progression in cancer patients, so a precise differential diagnosis was required [11].

In the context of emerging data and several meta-analyses on this manner [12,13,14], we present a series of 25 patients who presented at the Oncology Institute of Bucharest, Romania for either disease staging or treatment monitorization. 

## 2. Materials and Methods

### 2.1. Patient Selection

Our study included retrospective data from a total of 30 patients who underwent ^18^F-FDG PET-CT evaluations between July 2021 and March 2022, recorded at the Oncology Department of the Oncology Institute “Prof. Dr. Alexandru Trestioreanu”, Bucharest, Romania. The study was approved by the Institutional Ethical Committee. No specific informed consent form (ICF) was used because all patients signed the Institutional ICF giving consent for full use of their medical records for research purposes. 

Five patients were excluded due to extensive metastatic disease or hematological malignancies for which a clear differentiation between infiltrated lymph nodes and inflammatory lymph nodes was not possible. The patients who were in active chemotherapy had at least a treatment-free interval of three weeks before the examination.

### 2.2. PET/CT Acquisition Protocol

All ^18^F-FDG PET-CT evaluations were scanned using a 16-slice PET scanner (Discovery IQ, General Electric Healthcare), using the same protocol for all patients, following the ^18^F-FDG PET-CT procedure guidelines for tumor imaging provided by the European Association of Nuclear Medicine (EANM) [15]. The protocol included the administration of a diluted oral contrast agent and the injection of a dose of 2.5–3 MBq/Kg ^18^F-FDG (±10%), after a period of 6 h of fasting (with a target value for blood glucose of 70–160 mg/dL), followed by an uptake period of 60 ± 5 min. PET-CT examinations were acquired using the following settings: CT for attenuation correction: 140 keV and 30–130 mA Smart mA. Transverse images were reconstructed using filtered back projection with an attenuation kernel, slice thickness of 2.5 mm, and interval of 3.26 mm. PET scans were performed tridimensionally, with a scan time of 2–2.5 min/frame. Images were reconstructed using Ordered Subset Expectation Maximization (OSEM), 2 iterations and 10 subsets, and GE Q. Clear (General Electric Healthcare).

### 2.3. Image Interpretation

To account for the body weight fluctuations commonly seen in cancer and the overestimation of uptake in obese patients, radiotracer uptake was expressed as standardized uptake value (SUV) normalized by lean body mass (lbm). SUVlbm has long been recommended in practical use because ^18^F-FDG accumulation is minimal in white fat, and the percentage of this type of fat is high in obese patients, raising the possibility of SUV calculation errors [16,17]. In this study we used the maximum standardized uptake value (SUVmax) to assess the ^18^F-FDG assimilation in different lymph nodes.

The PET-CT findings were considered as part of post-vaccination status when there was a positive history for COVID-19 vaccination, associated with at least one of the following circumstances: metastatic malignant disease was controlled by current treatment line, considered stable or in partial response; cancer patient with no evidence of disease or in follow-up post-curative intent treatments; proof of benign/inflammatory histology on biopsy for highly suspicious cases of lymphatic metastasis (as in melanoma). A hyperactive lymph node was considered as adenopathy related to vaccination when it exceeded 10 mm in short axis diameter and had inflammatory characteristics: fatty hilum, oval shape, soft tissue density, and regular borders. 

We used the grading system established by Cohen at al. [18] for metabolically active lymph nodes post-vaccination, as follows: Grade 1 for SUVmax < 2.2, Grade 2 for SUVmax between 2.2–4, Grade 3 for SUVmax ≥ 4 in normal sized lymph nodes and Grade 4 for SUVmax ≥ 4 in enlarged lymph nodes.

### 2.4. Statistical Analysis

Analyzed variables are expressed in numbers and percentages, and mean for SUV as mean + SD (standard deviation). For correlation between vaccination time and PET-CT evaluation, a Spearman Rank Correlation test was used. All tests were conducted on www.evanmiller.org (accessed on 6 July 2022) [19] and SPSS Version 23 (IBM, SPSS, Inc., Chicago, IL, USA, 2015).

## 3. Results

The demographic and population characteristics are shown in Table 1 and Table 2. All patients had the vaccine injection in the deltoid muscle and received the second shot before the PET-CT examination. 

Most clinical findings related to vaccination were ipsilateral axillar adenopathies (96%), followed by ipsilateral cervical adenopathies and retropectoral adenopathies (both with 20%). One case also had contralateral axillar adenopathies, and two cases featured pulmonary hilum adenopathies. Some of the patients had more than one site of active adenopathies with the same SUV, thus being considered as a result of vaccination, without other pathological metabolic activity. 

The time interval between the vaccination and the PET-CT examination was correlated to the metabolic activity, this being emphasized in Table 3. Mean SUVmax was 3.51 with SD = 2.51. Median SUVmax was 2.85, with a minimum value of 0.95 and a maximum value of 10.57 (both calculated as descriptive statistics of SUVmax). As time passed from the vaccination moment, the SUV decreased considerably (Figure 1 and Figure 2). The trend towards the decreasing of SUV starts from week 4 post-vaccination and continues at 12 weeks post-vaccination. The lowest SUVmax registered in our study was 0.95, at 13 weeks post-vaccination, and maximum was 10.57, at 3 weeks. Pearson correlation test was R = −0.585 with *p =* 0.005, having a strong negative statistically significant correlation. This confirms the above theory that adenopathies’ uptake decreases in time after the booster shot. 

According to Cohen classification, there were nine patients (36%) with grade 1, seven patients (28%) grade 2, six patients (24%) grade 3 and three patients (12%) grade 4 metabolic activity.

There was no significant difference (*p =* 0.19) in the SUVlbm value of patients receiving Moderna mRNA-1273 vaccine (sample 1) vs. BNT162b2 mRNA Pfizer (sample 2) vaccine (Figure 3), irrespective of PET-CT timing.

After finding positive lymph nodes on a PET-CT scan performed for staging purposes, one patient had to undergo lymph node dissection. We obtained the pathological confirmation of inflammatory/benign axillar adenopathies in the context of COVID19 vaccination for this patient, who was originally suffering from malignant melanoma (Figure 4).

## 4. Discussion

Even though mRNA-1273 vaccine (Moderna) estimated about 16% of adverse reactions of axillary and cervical lymphadenopathy, in all grades, after the second dose, in all population age groups [20], recent published articles talk about a greater number of events, as most of the enlarged lymph nodes are clinically inapparent unless the patient is investigated through imaging for other reasons, such as breast cancer screening through mammogram/ultrasonography (US) or imagistic evaluation during treatment for malignant diseases through Positron Emission Tomography (PET-CT). 

A recent study by Cohen et al. [18] showed that the incidence of lymphadenopathy after BNT162b2 mRNA COVID-19 vaccine identified by ^18^F-FDG PET-CT, was up to 53.9% after the booster shot. Besides axillary adenopathies, which are easily visualized by US, it was discussed in various articles that the strong immune response to vaccines is observed on ^18^F-FDG PET-CT as an increased uptake in different and multiple lymph node territories, such as the mediastinum, the neck, the supraclavicular fossae and even the abdomen, following a vaccination in the deltoid muscle [21,22,23].

The mechanism behind such local adverse events is a pronounced immune response that seems to appear more often after the usage of mRNA-technology vaccines [24]. Nevertheless, this type of immune reaction has long been discussed in the literature and was formerly seen in cancer patients after the Influenza [25] or the Human Papilloma Virus (HPV) [26] vaccination on PET-CT examinations. The technical explanation for this appearance is the use of ^18^F-FDG, which is a glucose analogue that is captured in metabolically active glucose-using cells. Therefore, a differential diagnosis between inflammation/infection, neoplasia and immune activity post-vaccination is difficult to make. 

To raise awareness and address the need of recommendations for practice, McIntosh et al. [21] created an examination algorithm and a set of general recommendations for evaluating cancer patients with ^18^F-FDG PET-CT after a COVID-19 vaccine. Ideally, the vaccine should be made in the contralateral arm of the malignancy site, and the examination should be performed after four weeks following vaccination. If a nodal uptake is clinically relevant, another PET-CT should be performed 2–6 weeks afterwards to check for resolution and, in the case of a persistent uptake, an ultrasound sampling may be necessary for documentation of malignancy. If the suspicion is very high, a US core needle aspiration is mandatory. Expert radiologists from MD Anderson, TX, increased the period between the vaccination and the imaging examinations for oncological patients to 6 weeks [27]. If this is not possible, they recommend a baseline PET-CT or CT scan before vaccination. Other imagistic methods for lymph node staging, lymphoscintigraphy for example, may also be influenced by vaccination [28,29,30,31]. This comes in addition to the Society of Breast Imaging (SBI)’s recommendations, from Canada [32]. It is crucial now to obtain medical history, such as the vaccination status, the date of vaccination and injection site. They recommend that if a positive history of vaccination within 4–6 weeks of the sonography exists, following up closely in the next 12 weeks is advised. If the axillary adenopathy is still present, a biopsy is strongly recommended [32].

The question remains whether the six-week time interval is enough to ensure that there is no interference between vaccination status and the oncological disease, and if so, should we perform biopsies on all the hypermetabolic adenopathies found on PET-CT if six weeks post-vaccination have passed? 

The purpose of our retrospective trial is to shed some light on the actual context of the COVID19 pandemic and its implications in the oncological follow-up, and to draw a conclusion regarding the optimal timing of PET-CT evaluation after COVID-19 vaccination. Oncological patients are the vast majority who undergo a PET-CT examination [14]. To our knowledge, no other case series from Romania were published to this date and, in our opinion, center-oriented experience is helpful. 

Cohen et al. [18] established a grading system for metabolically active lymph nodes post-vaccination as follows: Grade 1 for SUVmax < 2.2, Grade 2 for SUVmax between 2.2–4, Grade 3 for SUVmax ≥ 4 in normal sized lymph nodes and Grade 4 for SUVmax ≥ 4 in enlarged lymph nodes. If we were to use the same grading system, we observe that four weeks after vaccination, the ^18^FDG uptake decreases to grade 2 and after 12 weeks it continues to decrease to grade 1. A higher uptake is seen in the first three weeks post-vaccination, which is an anticipated finding. In the Cohen et al. study, the persistence of a medium- to high-grade ^18^FDG uptake is seen in the first six days and washes out after 20 days, with only 7% of patients who still present Grade 3–4 SUVmax after three weeks. This also falls within our findings. 

El-Sayed et al., in their retrospective study of 204 vaccinated patients eligible for the analysis, found the six-week period post-vaccination showed a persistence of radiotracer activity of mean SUV 1.6 in women and 0.9 in men, with a persistence of up to 10 weeks in a lower percent. In the authors’ opinion, this time period of 1.5 months is representative enough to avoid acquiring a PET-CT scan, even though 14.5% of the study population still presented with significant metabolically active lymph nodes post-vaccination even after this period [12]. An important comment should be made noting the fact that the trial was conducted in England and, at that time, only one dose of vaccine had been administered, as per national protocols. We can assume, as seen in other analyses on the manner, that a second dose will provide a more sustained metabolic activity, with higher SUVs and longer periods of “washout” [18,33]. 

In another retrospective analysis, summing 140 patients with positive vaccination history and PET-CT acquisitions, only 54% had reactive metabolically active ipsilateral axillary lymph nodes overall. After a period of 28 days (four weeks), 36% of patients still presented a high mean SUVlbm of 3.9. Unfortunately, a strong correlation between the days that had passed after vaccination and the ^18^F-FDG uptake failed to be proven statistically [33], probably because of the population heterogeneity. 

It is still unclear whether 4–6 weeks post-vaccination is enough to ensure a low uptake of ^18^F-FDG on PET-CT. Tightly defining this time interval may create more difficulties for cancer patients and healthcare providers. 

The limitations of this study are related to the small sample size, with awareness that the number of patients is not sufficient for changing of practice, but our results are consistent with the meta-analyses and the case series previously mentioned in literature. A larger prospective study should be conducted to validate our results. 

## 5. Conclusions

In summary, revising the literature and considering our experience as a reference Oncological Center in Romania, we consider that the minimum interval of time to pass between an mRNA vaccine and a PET-CT examination should be more than six weeks. A detailed patient history should be obtained regarding vaccination status, including vaccines for Influenza, HPV or Hepatitis B. Following the Radiologist’s and Breast Cancer Society’s recommendations, the patient should have the immunization given in the contralateral side of the malignant tumor, if applicable, or have the baseline stabilization or reevaluation of the active cancer before having any vaccines [18,27,32]. 

## Figures and Tables

**Figure 1 diagnostics-12-02163-f001:**
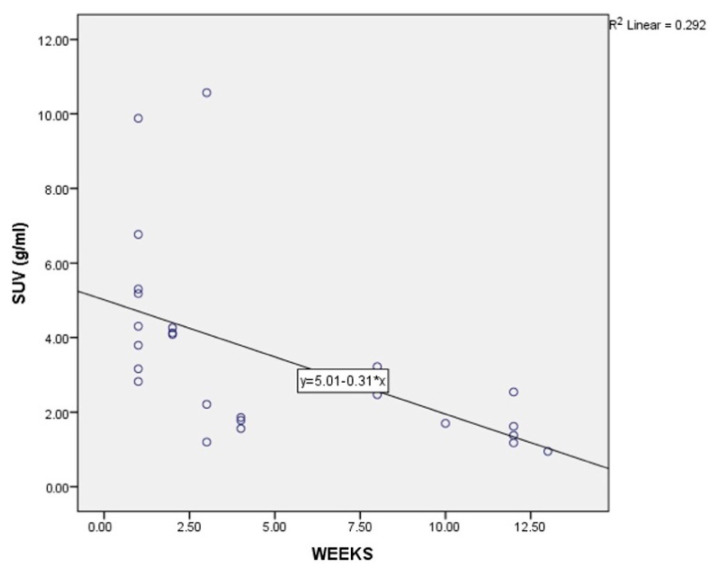
Correlation of SUVg/mL and time between vaccination and PET-CT examination; RHO = −0.54, *p* = 0.005.

**Figure 2 diagnostics-12-02163-f002:**
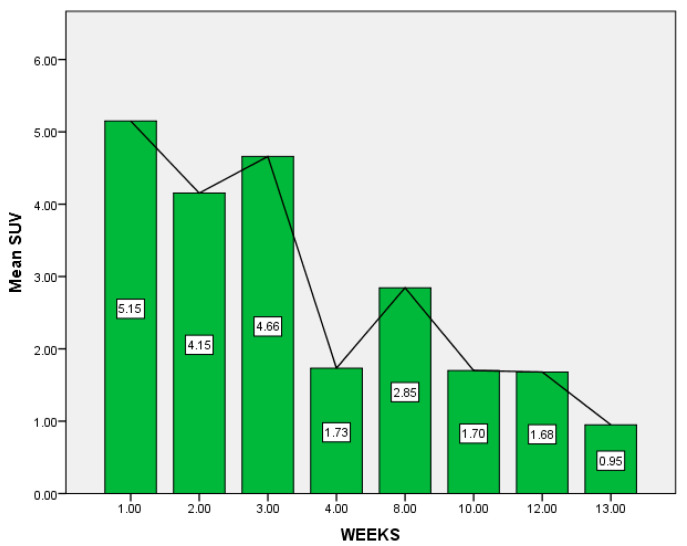
Variation of mean SUVg/mL according to time between vaccination and PET-CT examination in weeks.

**Figure 3 diagnostics-12-02163-f003:**
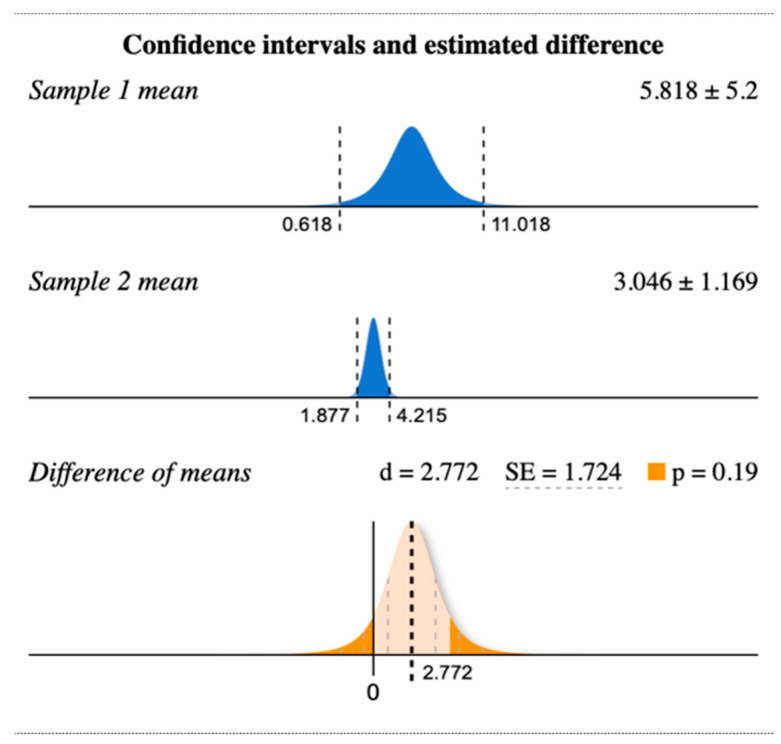
Difference of mean SUV of patients with Moderna mRNA1273 (sample 1) and mean SUV of patients with Pfizer BNT162b2 mRNA (sample 2). (www.evamiller.org).

**Figure 4 diagnostics-12-02163-f004:**
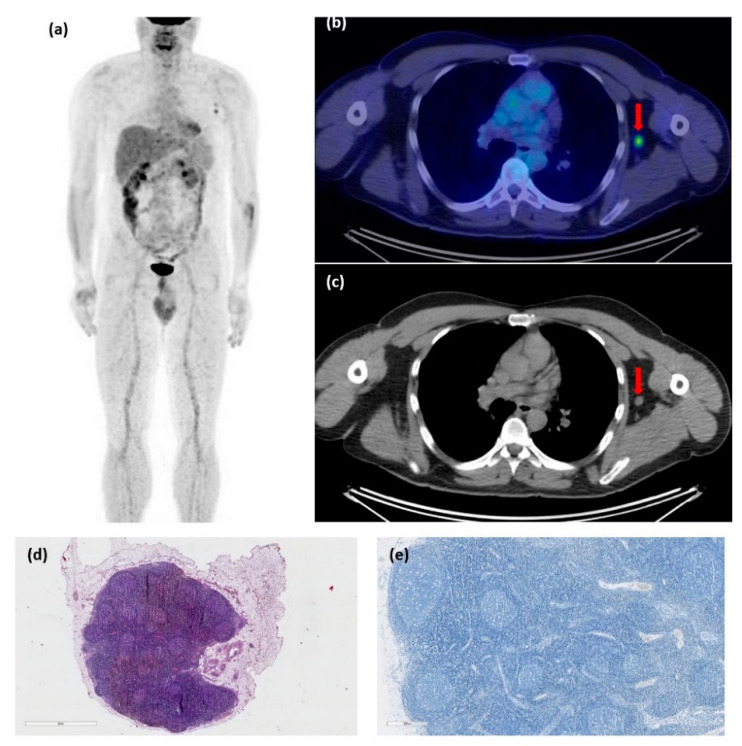
48-year-old male patient, diagnosed with malignant melanoma, pT3b pN0, follow-up scan. Image (**a**) shows the maximum intensity projection of the patient’s scan. Images (**b**,**c**) show a metabolically active axillary lymph node, on the same side as the mRNA vaccine, taken one week prior to the PET-CT examination, raising the suspicion of disease progression (marked with an arrow). A biopsy was performed, showing reactive changes in the lymphatic tissue (Hematoxylin and Eosin staining (**d**) and Melan-A immunostaining (**e**)).

**Table 1 diagnostics-12-02163-t001:** Patient characteristics. Abbreviations: CUP—cancer of unknown primary.

Characteristic	Number	Percentage
** *Gender* **		
Female	21	84%
Male	4	16%
Age		
<65	21	84%
>65	4	16%
** *Vaccine type* **		
Moderna	4	16%
Pfizer	15	60%
Unknown	6	24%
** *Diagnosis* **		
Breast cancer	6	24%
Lung cancer	2	8%
Colo-rectal cancer	8	32%
Melanoma	3	12%
CUP	1	4%
Head and Neck	1	4%
Ovarian cancer	2	8%
Sarcoma	1	4%
Pancreatic cancer	1	4%

**Table 2 diagnostics-12-02163-t002:** Reported findings on PET-CT examinations, according to the lymphatic region.

Lymphatic Region	Number	Percentage
*Ipsilateral axillar adenopathies*	24	96%
*Ipsilateral cervical adenopathies*	5	20%
*Ipsilateral retropectoral adenopathies*	5	20%
*Pulmonary hilum adenopathies*	2	8%

**Table 3 diagnostics-12-02163-t003:** Maximum value of SUVmax in concordance with the time interval between vaccination and PET-CT scan.

No. Weeks Between Vaccination and PET-CT	No. Patients	Maximum Value of SUV Max (lbm)
*1*	8	9.88
*2*	3	4.26
*3*	3	10.57
*4*	3	1.86
*8*	2	3.22
*10*	1	1.7
*12*	4	2.54
*13*	1	0.95

## Data Availability

Not applicable.

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
