# Peer review of "Understanding COVID Vaccination and Its Implication in Cancer Patients’ Imaging of Lymph Nodes by PET-CT"

_diagnostics, 2022, doi:10.3390/diagnostics12092163_

Round 1
Reviewer 1 Report
Novel analysis, clear methods and results
English could be improved ( construction of certain phrases could be improved). I suggest revision from an English mother-tongue person).
Author Response
Thank you for your review. Please see attached the revised version.

Reviewer 2 Report
The authors studied the influence of the time between vaccination and PET-CT scan on the diagnosis. Although the sample size was not big, the authors found that the minimum interval of mRNA vaccine administration and PET-CT examination should be longer than 6 weeks. There are some deficiencies that need to be corrected as listed below:
1. What is “SUVmax” in the abstract and main text?
2. How did the authors calculate the “Mean SUVmax”, “Median SUVmax”? For the “Median SUVmax was 2.82”, I did not find the value (“2.82”) in the Table 3.
3. What is the unit of SUV in Fig. 1.
4. There is a typo of the number of Fig. 2. What are the labels for x axis and y axis?
5. When the 18F-FDG was injected? Will the injection timepoint influence?
6. What is the influence of different mRNA vaccines on the optimal examination timepoint?
Author Response
Thank you for your review. Please see also attachment for updated version.
Point 1: What is “SUVmax” in the abstract and main text?
In this study we used the maximum standardized uptake value (SUVmax), to asses the 18F-FDG assimilation in different lymph nodes.
Point 2: How did the authors calculate the “Mean SUVmax”, “Median SUVmax”? For the “Median SUVmax was 2.82”, I did not find the value (“2.82”) in the Table 3.
Mean and Median SUVmax were both calculated as descriptive statisctics of SUVmax. A correction has been made to the table 3 and Figure 2 now includes value 2.82.
Point 3: What is the unit of SUV in Fig. 1.
The unit is g/ml.
Point 4: There is a typo of the number of Fig. 2. What are the labels for x axis and y axis?
The graphic was modified. X axis is WEEKS and y axis is mean SUV
Point 5: When the 18F-FDG was injected? Will the injection timepoint influence?
Please see text reference 135-138: The protocol included the administration of a diluted oral contrast agent and the injection of a dose of 2.5-3 MBq/Kg 18F-FDG (±10%), after a period of 6 hours of fasting (with a target value for blood glucose of 70 – 160 mg/dl), followed by an uptake period of 60 ±5 min.
The injection timepoint will not influence, because the difference between patients is small.
Point 6: What is the influence of different mRNA vaccines on the optimal examination timepoint?
Please see text reference 257-259: Based on our analysis, there was no significant difference (p=0.19) in the SUVlbm value of patients receiving Moderna mRNA-1273 vaccine vs. BNT162b2 mRNA Pfizer vaccine irrespectively of PET-CT timing.

Reviewer 3 Report
This work aimed to find an optimal timing between the vaccination and the PET-CT scan. The authors found a significant indirect correlation between SUVmax and the time passed between the vaccination and the PET CT. This work indicated that the minimum interval of time to pass between an mRNA vaccine and a PET-CT, should be of more than 6 weeks. In general, the structure of this manuscript is logically built. Good quality graphics accompanies the text. In my opinion, this work falls within the scope of the journal and this manuscript can be accepted after minor revisions.
1. In the section Introduction, a particular interest in oncology is represented by the immunological response at the site of injection and in the reactive lymphoid tissue surrounding it. More information about the immunity mechanism should be introduced herein. The relevant literatures have been listed below.
https://doi.org/10.1016/j.phymed.2021.153712
https://doi.org/10.1016/j.foodchem.2020.127933
2. The second figure should be "Figure 2", not "Figure 1".
Author Response
Thank you for your review. Please see attached the revised and updated version.

Round 2
Reviewer 2 Report
The authors need to further revise the manuscript carefully or can not be accepted.
1. The authors said they have done the following: "A correction has been made to table 3 and Figure 2 now includes value 2.82." But I did not see the correction.
2. There are still two "Figure 1".
3. The authors claimed, "The injection timepoint will not influence, because the difference between patients is small."
If the injection timepoint difference is 1 month or 1 year, is there still no difference?
Author Response
Thank you once again for taking the time to review our article. Below are our answers to your suggestions:
- The authors said they have done the following: "A correction has been made to table 3 and Figure 2 now includes value 2.82." But I did not see the correction.
The correction made to Table 3, mentioned in our previous report, is that we have erased the values for mean and median SUVmax, as they were not detrimental for the table and we considered them as misleading in that context. Thankfully, we also discovered that there was a typo in our manuscript, the correct value for median SUVmax being 2.85 and not 2.82. This value (2.85) can be found in Figure 2.
- There are still two "Figure 1".
We have performed the necessary changes.
- The authors claimed, "The injection timepoint will not influence, because the difference between patients is small."
If the injection timepoint difference is 1 month or 1 year, is there still no difference?
We think there was a misunderstanding regarding this point. We believed that “the injection timepoint” from your previous question (“When the 18F-FDG was injected? Will the injection timepoint influence?”) referred to the administration of 18F-FDG, not to the vaccination process, this being the reason why we explained the 18F-FDG administration and affirmed that the 60 ±5 minutes uptake interval would not influence our statistics.
If “the injection timepoint” you have mentioned referred to the vaccination timepoint, we have detailed the way the metabolic activity in the interested lymph nodes varied in correlation with the interval between immunization and PET/CT scan in Table 3 and lines 178-187:
“The time interval between the vaccination and the PET-CT examination was correlated to the metabolic activity, this being emphasized in Table 3. Mean SUVmax was 3.51 with SD=2.51. Median SUVmax was 2.82, with a minimum value of 0.95 and a maximum value of 10.57 (both calculated as descriptive statistics of SUVmax). As time passed from the vaccination moment, the SUV decreased considerably (Figure 1 and 2). The trend towards the decreasing of SUV starts from week 4 post-vaccination and continues at 12 weeks post-vaccination. The lowest SUVmax registered in our study was 0.95, at 13 weeks post-vaccination, and maximum was 10.57, at 3 weeks. Pearson correlation test was R = -0.585 with p=0.005, having a strong negative statistically significant correlation. This confirms the above theory that adenopathies’ uptake decreases in time after the booster shot.”
If you still consider that further changes still need to be made to our manuscript, please let us know.
